# OpenReview forum: "SteeringSafety: A Systematic Safety Evaluation Framework of Representation Steering in LLMs"
_ICLR.cc/2026/Conference — Submitted to ICLR 2026_

### Official Review · Reviewer_T7in · 2025-10-31

**Soundness:** 4
**Presentation:** 3
**Contribution:** 3
**Rating:** 8
**Confidence:** 3

**Summary:**

The paper outlines a systematic framework for the evaluation of representation steering methods across a variety of safety perspectives, to facilitate the performance on the selected safety perspectives and side effects on the other perspectives. The proposed framework, SteeringSafety, comprises modularized building blocks for representation steering methods such that existing methods can be represented as a combination of these blocks. This modularized breakdown enables the construction of novel representation steering methods by swapping out any of the modules. Besides this framework, the key result in this paper is that strong steering performance depends on the selected steering method, model and safety perspective.

**Strengths:**

The key contribution of this work is the framework, SteeringSafety. This enables the modularization of representation steering methods and the systematic evaluation of these methods on nine safety perspectives. The safety perspectives are drawn from literature, and suitable datasets and metrics are used for evaluation of methods on each perspective. This modular framework could be very useful for future development of representation steering methods and their subsequent testing. Additionally, the standardization of the tests would enable more consistent benchmarking in the future. The other notable aspect of this framework is the evaluation of entanglement on safety perspectives other than the ones the LLMs are aligned to.

I would also like to commend the authors on the general readability of the manuscript. They have done an admirable job of exemplifying the safety perspectives through the infographic in Figure 1, and opted to evaluate the methods on representation steering methods based on datasets and metrics used in existing literature. Furthermore, the breakdown of existing steering methods in terms of the defined modules (see Table 1) serves as a sound justification for the proposed modularization.

**Weaknesses:**

There are several misleading/confusing statements in this manuscript that need to be clarified prior to publication. The primary point of confusion is the restriction of measuring steering effectiveness on three main perspective axes (line 67) – it is not evident why only three perspectives are selected and which three perspectives the authors are referring to.

In addition to this, the description and mathematical notation used in Section 3.1.3 is quite confusing – the mathematical form of $\nu^{\prime}$ is incomplete as there is seemingly an additional projection operation not included in the mathematical expression.

I have listed my remaining questions and suggestions in the subsequent section.

**Questions:**

**Questions:**

1.	Why does the metric for effectiveness (Eq. 1) include a normalization by $1 – y_b$? Wouldn’t this result in effectiveness having a different range than entanglement?
2.	Line 267 states that “we search from the 25th to the 80th quantile of the layers” but line 268 states that “prior work has shown steering is more effective in the middle layers.”

  a.	Are these quantiles based on the values or position of the layers?

  b.	If it is based on values, what does the latter statement mean?
3.	The description of directional ablation in Section 3.1.3 (line 286) appears to modify the activations in a direction orthogonally to $\ni$, which contrasts with activation ablation. Is this correct? If yes, how can you reconcile the difference in direction ablations across the two approaches.
4.	Is line 343 supposed to read “decreasing harmfulness” since lower values for harmfulness would be desirable?


**Suggestions:**

Section 5 should be moved to earlier in the paper as it is important to establish the current landscape of evaluation of steering methods on safety perspectives. This would better establish the utility of the proposed framework.

---

> ### Author Response · Authors · 2025-11-21
>
> Thank you for your valuable insights and feedback. We’re especially pleased that you appreciated the readability and clarity of our work through our figures and tables and SteeringSafety’s potential for future work. We address your concerns and suggestions below:
>
> ---
>
> ### `W1: Misleading statement about three main perspective axes (line 67)`
>
> Thank you for pointing this out. We apologize for the confusion. The three perspectives we steer on are: harmfulness, bias, and hallucination (lines 341-348). We selected these because they (1) are most aligned with existing steering literature, allowing direct comparison with prior work, and (2) have sufficiently large datasets to support robust direction extraction and evaluation. Several other perspectives have not been steered in prior work or have limited data (e.g., DarkBench behaviors contain only 110 prompts total) , which is insufficient for reliable steering vector extraction and evaluation.
>
> However, we recognize the value of steering additional perspectives beyond the three original axes. **We are running experiments to extend our evaluations to an additional axes** and will report preliminary results soon.
>
> ---
>
> ### `W2 & Q3: Confusing description and incomplete mathematical notation in Section 3.1.3`
>
> We acknowledge that Section 3.1.3 is unclear. To clarify: there are two distinct operations:
> 1. Directional Ablation (used in DIM): Removes the component along a direction via orthogonalization.
> 2. Directional Ablation + Affine (used in ACE): Applies orthogonalization then adds an affine shift.
>
> We will separate these operations clearly with distinct equations in the revision.
>
>
> ---
>
> ### `Q1: Effectiveness metric normalization creates different range than entanglement`
> We normalize effectiveness (Equation 1) to ensure equal weighting across perspectives when aggregating results, preventing perspectives with naturally larger effect sizes from dominating the average. This normalization is necessary because steering effectiveness varies dramatically across perspectives (e.g., harmfulness shows 5-10× larger changes than bias).
> For entanglement, we deliberately use absolute (non-normalized) changes because entanglement effects are often small and near-zero, making relative changes misleading (e.g., 0.2%→1% would appear as 500% increase despite being noise). Additionally, we want to identify the largest-magnitude side effects regardless of baseline rates. This design choice allows us to focus on practically significant entanglement rather than proportional changes. We will add this clarification to Section 3.2.
>
> ---
>
> ### `Q2: Confusion about layer quantiles (lines 267-268) - based on values or position?`
>
> The quantiles are based on the positions of the layers, i.e., if there are L layers, we search from int(L/4) rounded down to layer int(4L/5) rounded up. This searches the middle layers targeted in prior work. We will include a clarification in a newer version.
>
> ---
>
> ### `Q4: Line 343 should read "decreasing harmfulness"?`
>
> This is intentional, not a typo. Current representation steering literature focuses on jailbreaking (i.e., increasing harmfulness), and we adopt this convention to maintain consistency with prior work. This allows our benchmark to assess both (1) how steering interventions are currently used to bypass safety measures, and (2) how refusal mechanisms interact with other model behaviors. Importantly, our entanglement results reveal that refusal steering consistently affects social behaviors like sycophancy, mirroring findings from RLHF work on safety alignment. We will add explicit motivation for this design choice in the revision.
>
> ---
> ### `Moving the Related Work`
> We appreciate this suggestion. We recognize the value of establishing the research landscape earlier and will reorganize the related work earlier to better motivate the need for SteeringSafety.

---

> > ### Author Response · Authors · 2025-11-23
> >
> > ### Follow-up: Experiments in Progress
> >
> > To address the concerns raised, we are running the following experiments for our revision:
> >
> > **1. Extended Steering Perspectives (W1)**
> >
> > - Running experiments to extend our evaluations beyond the three original axes (harmfulness, bias, hallucination)
> > - Testing steering on additional perspectives where data permits robust direction extraction and evaluation
> > - Preliminary results will be reported soon
> >
> > Additionally, we will make the following updates to improve clarity and reproducibility:
> >
> > - Clarified Mathematical Notation (W2 & Q3): Separating Directional Ablation and Directional Ablation + Affine into distinct equations; providing complete mathematical expressions; restructuring Section 3.1.3
> > - Enhanced Metric Documentation (Q1): Adding detailed clarification in Section 3.2 explaining normalization rationale for effectiveness vs. entanglement metrics
> > - Layer Selection Clarification (Q2): Clarifying quantiles are position-based; adding explicit formula (int(L/4) to int(4L/5))
> > - Design Choice Motivation (Q4): Adding explicit motivation for increasing harmfulness focus and its connection to jailbreaking assessment
> > - Paper Reorganization: Moving Section 5 (Related Work) earlier to better establish research landscape
> >
> > We will share preliminary results from experiment 1 as soon as they are available and will incorporate all updates into our revised submission.

---

### Official Review · Reviewer_cyiT · 2025-11-01

**Soundness:** 3
**Presentation:** 3
**Contribution:** 3
**Rating:** 6
**Confidence:** 2

**Summary:**

This paper introduces STEERINGSAFETY, a comprehensive framework for evaluating representation steering methods in LLMs across several safety perspectives using subsets from 17 datasets. It aims to systematically assess both the effectiveness and unintended side effects of steering interventions.

**Strengths:**

- Unifies five popular steering methods under the same benchmark, allowing for the evaluation of effectiveness and the side effects of these methods.
- Empirically, the results provide valuable insights into entanglement, showing that effectiveness and safety trade-offs vary across models and methods, which is a key contribution to alignment research.
- The paper is also well-situated within related work and connects systematically to ongoing research in the field.

**Weaknesses:**

- While the framework claims modularity and standardization, it doesn’t provide runtime cost  or inference-time overhead details.
- Effectiveness and entanglement are reported using scaled averages (in the main paper), which can hide nuanced behavior shifts. The results in the appendix seem more informative.
- For some categories, the number of used prompts seem a bit low.
- The ethics statement is minimal (“our goal is to improve safety”). The reliance on LLM-as-a-judge should also be discussed given how it can mask some biases in the evaluation.

**Questions:**

- Can the authors put the definitions of the steering methods’ acronyms earlier in the paper?
- For categories with relatively few prompts (like social behaviors), how do you ensure statistical robustness in the evaluation?
- Given the reliance on GPT-4/ llama3 for scoring certain behaviors, how do you account for potential biases or inconsistencies in its judgments?

---

> ### Author Response · Authors · 2025-11-21
>
> Thank you for your valuable insights and feedback. We’re pleased that you appreciate our contributions to alignment research and our applicability to the developing representation steering field. We address your concerns below:
>
> ---
>
> ### `W1: Missing runtime cost and inference-time overhead details`
>
> We agree this information is important. Our experiments are computationally efficient: each full evaluation (direction generation, selection, application, and scoring across all datasets) takes 1-3 hours on a single GPU (A6000/A100/H100). The complete benchmark comprises 180 experiments (5 methods × 3 perspectives × 3 models × 3 settings, plus additional Qwen evaluations), which can be run in parallel across multiple GPUs. Total compute time ranges from 180-540 GPU-hours. We use API-based evaluation (OpenAI, Groq) and locally hosted LlamaGuard with vLLM for efficient scoring. We will include these computational details and estimated API costs in the revision.
>
> ---
>
> ### `W2: Scaled averages hide nuanced behavior shifts; appendix results more informative`
>
> We agree that more detailed analysis would strengthen the main paper. In our revision, we will include: (1) more intuitive figures showing per-perspective breakdowns, and (2) expanded discussion of the results, including average effectiveness and entanglement by perspective and the ratio between them. The appendix results will remain available for readers seeking granular details.
>
> ---
>
> ### `W3 & Q2: Low number of prompts for some categories`
>
> We acknowledge that some perspectives have fewer prompts, such as the DarkBench behaviors with 110 prompts total. To ensure robustness despite smaller sample sizes, we have conducted statistical significance tests for the per-model results (to be included in Appendix E of the revision). Our analysis applies appropriate multiple testing corrections and focuses on effect sizes rather than significance alone, ensuring that observed differences between steering methods are statistically reliable.
>
> ---
>
> ### `W4 & Q3: Minimal ethics statement; LLM-as-a-judge can mask evaluation biases`
>
> We agree that using LLMs in this way can mask biases. However, because all steering methods are evaluated under identical judging conditions, our relative comparisons (which are the focus of SteeringSafety) remain robust to systematic biases. We will adjust the ethics statement in the updated version to include discussion of our reliance on LLM-as-a-judge and our choice of models based on prior work for harmfulness [1], extrinsic hallucination [2], and the DarkBench [3] datasets. To further validate our findings, we will include human annotations on a representative subset in our revision, which will allow us to quantify inter-rater reliability between human and LLM judges.
>
> ---
>
> ### `Q1: Steering method acronym definitions should appear earlier`
>
> Thank you for pointing this out. We will move the acronyms earlier in the updated version.
>
> ---
>
> ### References
>
> [1] Andy Arditi, Oscar Obeso, Aaquib Syed, Daniel Paleka, Nina Panickssery, Wes Gurnee, Neel Nanda: Refusal in Language Models Is Mediated by a Single Direction. NeurIPS 2024
>
> [2] Yejin Bang, Ziwei Ji, Alan Schelten, Anthony Hartshorn, Tara Fowler, Cheng Zhang, Nicola Cancedda, Pascale Fung: HalluLens: LLM Hallucination Benchmark. ACL (1) 2025: 24128-24156
>
> [3] Esben Kran, Jord Nguyen, Akash Kundu, Sami Jawhar, Jinsuk Park, Mateusz Maria Jurewicz: DarkBench: Benchmarking Dark Patterns in Large Language Models. ICLR 2025

---

> ### Author Response · Authors · 2025-11-23
>
> ### Follow-up: Experiments in Progress
>
> To address the concerns raised, we are running the following experiments for our revision:
>
> **1. Human Annotation Study (W4 & Q3)**
>
> - Conducting human annotations on a representative subset of evaluation data
> - Quantifying inter-rater reliability between human annotators and LLM judges
> - We will focus more human annotation efforts on datasets with lower sample sizes to ensure their validity
> - This will validate our LLM-as-a-judge approach and address concerns about masked biases and sample sizes
>
> **2. Low Prompt Numbers (W3 & Q2)**
>
> - Conducting statistical significance tests for per-model results across all perspectives
> - Applying appropriate multiple testing corrections
> - Including these statistical analyses in Appendix E to validate findings for categories with fewer prompts
> - Investigating if larger datasets are available for these categories and will include these in our new experiments
>
> Additionally, we will make the following updates to improve clarity and reproducibility:
>
> - Computational Cost Analysis (W1): Documenting runtime costs, total compute time (180-540 GPU-hours), estimated API costs, and inference-time overhead details
> - Enhanced Results Presentation (W2): Adding intuitive per-perspective breakdown figures; expanding discussion of effectiveness and entanglement; reporting ratios between metrics
> - Expanded Ethics Statement (W4 & Q3): Updating to discuss LLM-as-a-judge reliance and model selection based on prior work
> - Improved Paper Organization (Q1): Moving steering method acronym definitions earlier in the paper
>
> We will share preliminary results from experiments 1-2 as soon as they are available and will incorporate all updates into our revised submission.

---

### Official Review · Reviewer_3DPr · 2025-11-03

**Soundness:** 3
**Presentation:** 3
**Contribution:** 3
**Rating:** 8
**Confidence:** 3

**Summary:**

In this paper, the authors introduce STEERINGSAFETY as a framework for evaluating how representation steering methods affect safety across multiple dimensions in LLMs. The paper points out that previous research has focused on improving performance or alignment with steering methods, but this work shows that such interventions can also introduce unintended safety side effects.

They systematically evaluate and compare representation steering methods across nine safety dimensions and 17 datasets, revealing unintended safety trade-offs. Their results show that steering outcomes depend on the specific combination of steering method and model, with some interventions improving some target behaviors but harming others.

**Strengths:**

* This paper offers a broad and systematic evaluation to assess representation steering methods across multiple safety dimensions (nine perspectives, 17 datasets).
* The authors provide consider diverse steering techniques (e.g., DIM, ACE, CAA, PCA, LAT) for their experiments.
* Novel safety metrics: The framework introduces Effectiveness and Entanglement metrics to assess both steering performance and side effects, which allow tradeoff analysis. However a comparison to related metrics used for safety assessment is missing.
* The paper reveals that steering effectiveness depends strongly on the combination of model & steering method. No single best method exists, and they discuss some method-specific insights, e.g. DIM offers strong performance but at the cost of safety trade-offs.
* The presented insights can help the interpretability community develop steering interventions with finer behavioral control and fewer unintended harms.

**Weaknesses:**

* The framework identifies safety tradeoffs  but does not clearly explain why the studied methods are related to specific harms. The underlying causal mechanisms remain often unclear to the reader.
* The evaluation relies on LLM-based scorers, which can introduce bias and reduce interpretability of the results. Wouldn't evaluation with the help of human annotators for at least a subset of results provide more insights?
* Although the benchmark covers nine safety perspectives, it leaves out some very relevant safety dimensions such as adversarial robustness, long-context reasoning stability, etc.
* The dynamic testing approach uses only small data subsets (20% of each dataset), which might cause some unintended side effects. Also details on selection of the subsets is missing?
* Some methodological details, e.g. how layer selection impacts with steering strength, are not discussed in detail > can result in issues with reproducibility.
* The figures summarize results across models and methods but are are dense and difficult to interpret without further breakdowns

**Questions:**

1. Have you considered including ablation studiesto show how results change with different parameter/steering settings?

2. Could you provide more details on how LLM judges like GPT-4o or LlamaGuard were prompted across datasets to ensure consistent scoring?

3. Could you include finer-grained results or plots to help interpret the tradeoffs more clearly?

---

> ### Author Response · Authors · 2025-11-21
>
> Thank you for your valuable insights and feedback. We’re pleased that you appreciated the broadness of SteeringSafety and its applicability to develop better steering interventions. We address your concerns below:
>
> ---
>
> ### `W1: Unclear causal mechanisms linking methods to specific harms`
>
> We highlight that SteeringSafety is primarily an evaluation framework for assessing specific harms based around seventeen sub-perspectives, almost all of which are uncharacterized in prior work. Therefore, the causal mechanisms underlying these novel harms are presently unknown. Our results reveal strong variation across methods and models which suggests that discovering causal mechanisms between these perspectives is an important direction for future work.
>
> ---
>
> ### `W2 & Q2: LLM-based scorers introduce bias and reduce interpretability`
>
> We selected LLM judges that match those used in the original dataset papers to ensure consistency with established benchmarks. Because all methods are evaluated under identical judging conditions, our relative comparisons remain robust to systematic biases. To further validate our findings, we will include human annotations on a representative subset in our revision, which will allow us to quantify inter-rater reliability between human and LLM judges.
>
> ---
>
> ### `W3: Missing safety dimensions (adversarial robustness, long-context reasoning stability, etc.)`
>
> **Adversarial Robustness**:  SteeringSafety evaluates steering of harmfulness by increasing harmfulness. This is equivalent to jailbreaking the target model and directly assesses the adversarial robustness of the tested models. Currently, as seen in the results figures in Section E.1, we find that jailbreaking through steering is easier to accomplish than decreasing bias or hallucination, suggesting the need for more robust post-training methods to enforce refusal.
>
> **Long-context Reasoning**: We thank the reviewer for this suggestion. We believe long-context reasoning risks exceeding the context windows of our tested models (eg. Gemma 2, 8192 token context window) and may be too similar to our reasoning benchmarks (GPQA and ARC-C). However, we recognize reasoning stability is a meaningful metric for us to assess. Therefore, we are adding a perplexity evaluation for our steered models on a long corpus (eg. the Pile) and will report back preliminary results.
>
> We would also like to highlight that we intend SteeringSafety to serve as a living benchmark. Just as we hope to expand our benchmark to novel representation steering work in new modalities (e.g., vision), we also plan to add additional new safety dimensions when relevant benchmarks are released.
>
> ---
>
> ### `W4: Dynamic testing uses small data subsets (20%) without clear selection details`
>
> We used 20% of each dataset to manage computational cost while maintaining statistically meaningful sample sizes. The exact split logic is documented and released in our public code, and the revised version will include a table summarizing dataset sizes and splits. We will also release the full processed HuggingFace dataset to ensure full reproducibility.
>
> ---
>
> ### `W5 & Q2: Missing methodological details affecting reproducibility (layer selection, steering strength, LLM judge prompting)`
>
> All details of our pipeline, including layer selection, steering strength, and judge prompting, are contained in our public code release. An upcoming revision will include an appendix summarizing these choices for clarity and ease of reference.
>
> ---
>
> ### `W6 & Q3: Dense figures difficult to interpret without finer-grained breakdowns`
>
> We acknowledge that the results are not analyzed in as much detail and as clearly as they should be. In response, we have redesigned our evaluation section to include more figures and present more clear analysis. The revision we are submitting will include updated figures.
>
> ---
>
> ### `Q1: Ablation studies on different parameter/steering settings`
>
> Many prior works, including the original CAA work and others, have tested their steering methodologies with more diverse parameters such as different layers and steering coefficients. While this is a potential source of interest for investigating entanglement, our main focus is also to establish steering effectiveness on the best grid-searched layer. Each additional parameter tested would multiply our computational load proportionally. Therefore, we leave these ablation studies to further works focusing more specifically on single methodologies.

---

> ### Author Response · Authors · 2025-11-23
>
> ### Follow-up: Experiments in Progress
>
> To address the concerns raised, we are running the following experiments for our revision:
>
> **1. Human Annotation Study (W2 & Q2)**
>
> - Conducting human annotations on a representative subset of our evaluation data
> - Quantifying inter-rater reliability between human annotators and LLM judges (GPT-4o, LlamaGuard)
> - This will validate our LLM-based scoring approach and address concerns about potential bias
>
> **2. Long-Context Reasoning Stability Evaluation (W3)**
>
> - Adding perplexity evaluation on steered models using a long corpus exceeding tens of thousands of tokens
> - This will assess stability relative to the original model as an additional safety dimension beyond our current benchmarks
> - Preliminary results will be included in the revision
> - Evaluate feasibility of running Llama/Qwen models on a subset of long-context benchmarks and include relevant results if possible
>
> Additionally, we will make the following updates to improve clarity and reproducibility:
>
> - Enhanced Documentation (W4, W5, Q2): Adding appendix table with dataset sizes and split details; including comprehensive methodological appendix for layer selection, steering strength, and judge prompting; releasing full processed HuggingFace dataset
> - Improved Results Presentation (W6 & Q3): Redesigning evaluation section with clearer figures; adding finer-grained breakdowns for interpreting safety tradeoffs; providing more detailed analysis
> - Early Acronym Definitions (implied): Moving steering method acronyms earlier in the paper
>
> We will share preliminary results from experiments 1-2 as soon as they are available and will incorporate all updates into our revised submission.

---

### Official Review · Reviewer_NFy8 · 2025-11-04

**Soundness:** 1
**Presentation:** 3
**Contribution:** 2
**Rating:** 2
**Confidence:** 2

**Summary:**

This paper offers a framework for safety assessment for models that looks at safety across seven dimensions, operationalizing each with a measure. The paper further assesses safety of several large models using this framework and applying steering methods that intervene directly on neural network activations, requiring no training. The paper offers the assessment toolchain as a contribution. A major finding is that safety properties are "entangled" both in this assessment and in prior work, in that improving performance on one operationalization of safety may harm performance on another.

**Strengths:**

+ The notion that multiple operationalizations of safety may be necessary to make sense of safety claims is important, as is the notion that improvement in one sense may harm performance in another. This is reminiscent of various "no-go" theorems (although there is no formalism offered here, and there could not be because safety claims are epistemically non-falsifiable) that demonstrate e.g. the no-free-lunch theorem or results from Chouldechova (resp. Kleinberg) that it will be impossible to square different operationalizations of bias except under specific theoretical circumstances (e.g., a perfect predictor or already-equalized feature incidence per class stratum in the dataset) or even Kleinberg's clustering impossibility result.
+ The paper undertakes a substantial level of effort with regard to benchmarking and evaluation.

**Weaknesses:**

- It is never made clear how the seven framework dimensions were selected or why they were operationalized using the specific measures chosen. While each is a topic that has been studied in much prior work, claims that the framework is "comprehensive" must be supported by arguments for the completeness/soundness of the evaluation with regard to some exogenously developed notion of safety hazards/risks not tolerated. Indeed, the "framework" reads more like a wish list, making it hard to evaluate its quality against any other proffered "framework", like the NIST AI RMF or any company's internal framework.
- The framework is offered as a contribution, but there is no effort in the paper to evaluate the framework other than to apply it. It is not possible to understand if the results obtained represent a purposive sample of safety benchmarking or if applying the framework in some application would reduce some kind of risk or avoid some defined hazard (this is the standard approach to evaluating safety in most disciplines, or there is some accepted proxy metric). The paper could, for example, argue in favor of this breakdown of safety issues to show that it covers the bulk of concerns in this literature (while a literature survey is probably a different paper, arguing for the "comprehensive" or "systematic" nature of the taxonomy requires analysis that is not given here).
- I think the paper suffers from a kind of category mistake in which "safety" is a property of the model-under-test and not of the model-in-use in some use case. If it is true (as the paper argues) that improving safety performance in some aspects worsens it in others, the next natural question is about whether "good enough" performance exists for some use case. But here, the models are considered only for the use case of "performance on a selected benchmark" and the paper in its current form does little to link the benchmark contexts to real-world, outside-the-lab use cases or to argue that the chosen benchmarks reflect those use cases well. Instead, a claim of safety is a claim about avoiding certain defined bad outcomes - how does the analysis approach here do when measured against that goal? One thing the framework can help with is mapping between the risks operationalized by the chosen benchmarks and the use cases in which the models-under-test might be applied; in doing so, the framework can help to navigate the tradeoff identified that "there is no universal best method that maximizes effectiveness while minimizing entanglement across all models" [384-386] (a claim so broad it cannot possibly be true).
- Many of the stated contributions in the intro and conclusion are not substantiated by the paper. In addition to the issues of framework comprehensiveness and usefulness mentioned above, there is no effort in the paper to explain why the assessments are "systematic" or "reproducible" or "reliable". These are empirical claims about the use of the framework to evaluate real applications; showing that it is possible to pose questions for which evaluations lead to quantitative results only matters if there is meaning to be made from the resulting quantifications. Here, I think the paper shows its greatest weakness, in that no such meaning is extracted from the many evaluations provided. There may be contribution within these evaluations, but that is not the way the paper presents itself.

**Questions:**

* Why is this framework "comprehensive" or "systematic", as the paper title and listed contributions claim?
* How would a user of this approach gain capacity to navigate the tradeoffs identified in the evaluations performed using the techniques in this work? (Either across dimensions in the framework or across models - and to what extent does each of those degrees of freedom matter?)
* Can the claims about safety evaluation be toned down such that the evaluation work in the paper supports them sufficiently? How?

---

> ### Author Response · Authors · 2025-11-21
>
> We appreciate your thoughtful engagement and would like to clarify the scope of our work.
>
> ## Clarifying the Scope: An Evaluation Framework for Steering Methods
>
> **Addressed: "This paper offers a framework for safety assessment for models"**
>
> Our paper introduces an **evaluation framework for representation steering methods**, not a normative framework for AI safety at large. As stated in our title and throughout the paper, we systematically assess how steering interventions affect behavior across multiple established safety benchmarks. We do not claim to define what "safety" means universally. Rather, we evaluate steering methods using widely-accepted operationalizations from prior work (BBQ for bias, TruthfulQA for hallucination, etc.).
>
> ---
>
> ## W1 & Q1: Framework Dimension Selection
>
> **Addressed: "It is never made clear how the seven framework dimensions were selected" + "Why is this framework 'comprehensive'?"**
>
> Our nine perspectives were selected based on three criteria:
>
> 1. **Established benchmarks exist** with validated evaluation metrics and sufficient data to support evaluation and steering
> 2. **Prior steering work** has demonstrated effects on these dimensions (citations on lines 343-438)
> 3. **Safety literature** identifies these as high-priority risk areas, such as the NIST AI RMF you reference.
>
> We use "systematic" to mean consistent evaluation protocols applied across all method-model combinations. If "comprehensive" is interpreted as "covering all possible safety dimensions," we acknowledge this is beyond our scope and will clarify our language. Our framework covers the perspectives most relevant to current representation steering research where robust evaluation is possible.
>
> ---
>
> ## W2: Evaluating the Framework's Utility
>
> **Addressed: "there is no effort in the paper to evaluate the framework"**
>
> The value of an evaluation framework lies in enabling systematic comparison and discovery. SteeringSafety demonstrates utility through:
>
> 1. **Novel empirical findings:** We reveal widespread entanglement effects, that steering one perspective consistently affects others, which was largely uncharacterized across previous steering methods
> 2. **Method comparison insights:** We show that effectiveness depends critically on method-model-perspective combinations, with no universally best approach
> 3. **Community infrastructure:** Our released code, standardized protocols, and benchmark suite enable reproducible comparison of future steering methods
>
> This follows the model of successful benchmarks in other domains where standardized evaluation infrastructure enables systematic progress.

---

> ### Author Response · Authors · 2025-11-21
>
> ## W3 & Q2: Connection to Use Cases and Navigating Tradeoffs
>
> **Addressed: "little to link the benchmark contexts to real-world, outside-the-lab use cases" + "How would a user navigate tradeoffs?"**
>
> Our work targets the representation steering research community, where controlled evaluation in standardized settings is essential for understanding method behavior before deployment. One of our key findings, that steering interventions exhibit strong entanglement, has direct implications for any real-world application: developers must account for unintended side effects when using steering methods.
>
> **Navigating tradeoffs:** Our framework reveals empirical tradeoffs but does not prescribe how to resolve them. We believe this is appropriate for an evaluation framework. Users can examine our results or test their own use cases to identify which method-model-perspective combinations best suit their specific goals. For example:
> * Researchers prioritizing harmfulness reduction can identify which methods achieve high effectiveness with minimal entanglement on their target model
> * Those focused on bias can compare relative performance across methods
> * Developers can assess whether conditional steering reduces entanglement for their use case
>
> We will add additional discussion on takeaways for practitioners in our revision.
>
> ## W4 & Q3: Substantiating Our Contributions
>
> **Addressed: "Many of the stated contributions in the intro and conclusion are not substantiated" + "Can claims be toned down?"**
>
> We believe our contributions are directly substantiated by our empirical results:
>
> - **"No method is universally effective"**: Demonstrated in Figure 2 and Section 4.2 with quantitative results across all method-model-perspective combinations
> - **"Strong entanglement exists"**: Shown in Figure 2 and discussed in Section 4.2 - steering one perspective consistently affects others
> - **"Conditional steering does not reliably remove entanglement"**: Demonstrated in Figures 8 and 11 with quantitative analysis
>
> These are empirical findings from our experiments, not normative claims about safety.
>
> If specific phrasing in our introduction or conclusion suggests we are defining universal safety principles rather than evaluating steering methods, we will revise for clarity. However, our core contributions are empirical and directly substantiated by our results. We will strengthen the presentation of these claims with additional figures, statistical testing, and expanded discussion in our revision.
>
> ---
>
> ## On Terminology: "Systematic," "Reproducible," and "Reliable"
>
> **Addressed: "there is no effort in the paper to explain why the assessments are 'systematic' or 'reproducible' or 'reliable'"**
>
> These terms describe properties of our evaluation methodology, which we can clarify more explicitly:
>
> - **Systematic:** Consistent evaluation protocols applied across 5 methods × 3 variants × 5 models × 17 datasets
> - **Reproducible:** Full code release, documented evaluation protocols, and processed datasets on HuggingFace
> - **Reliable:** Use of established benchmarks with validated metrics from prior work
>
> These characterize our evaluation process, and we will make this distinction more explicit throughout the paper.
>
> ---
>
> ## Summary
>
> Our paper introduces an evaluation framework for representation steering methods. Our contributions are:
>
> 1. **Methodological:** Standardized infrastructure for systematic steering method comparison
> 2. **Empirical:** Discovery of widespread entanglement effects and method-model-perspective dependencies
>
> These are directly substantiated by our experiments. We will revise any language that suggests we are proposing a normative safety framework rather than an evaluation framework for steering methods. Our goal is to enable systematic assessment of steering interventions, not to define safety universally.

---

### Author Response · Authors · 2025-11-27

## Summary of Revisions

We thank all reviewers for their thorough and constructive feedback. We are encouraged by the shared recognition of SteeringSafety's **systematic safety evaluation approach for representation steering methods**, **modular framework design**, **valuable empirical insights on safety entanglement across steering interventions**, and **new experimental results addressing reviewer concerns**. Below, we summarize reviewer endorsements, address remaining concerns, and highlight new experimental results.

In response to the feedback from the reviewers, we have made improvements in the paper, with corresponding revisions highlighted in yellow. We also highlight the manuscript’s attached anonymized code to facilitate reproducibility and future research.

---

### Reviewer Endorsements

We appreciate that reviewers acknowledged the following strengths of SteeringSafety:

**Comprehensive and systematic safety evaluation framework for steering methods**
- **3DPr**: *"broad and systematic evaluation to assess representation steering methods across multiple safety dimensions (nine perspectives, 17 datasets)"*
- **T7in**: *"This modular framework could be very useful for future development of representation steering methods and their subsequent testing. Additionally, the standardization of the tests would enable more consistent benchmarking in the future"*
- **cyiT**: *"Unifies five popular steering methods under the same benchmark, allowing for the evaluation of effectiveness and the side effects of these methods"*

**Novel empirical findings on safety entanglement**
- **3DPr**: *"Novel safety metrics: The framework introduces Effectiveness and Entanglement metrics to assess both steering performance and side effects, which allow tradeoff analysis"*
- **3DPr**: *"The paper reveals that steering effectiveness depends strongly on the combination of model & steering method. No single best method exists"*
- **cyiT**: *"Empirically, the results provide valuable insights into entanglement, showing that effectiveness and safety trade-offs vary across models and methods, which is a key contribution to alignment research"*

**Clear presentation and methodological rigor**
- **T7in**: *"I would also like to commend the authors on the general readability of the manuscript. They have done an admirable job of exemplifying the safety perspectives through the infographic in Figure 1"*
- **cyiT**: *"The paper is also well-situated within related work and connects systematically to ongoing research in the field"*

---

> ### Author Response · Authors · 2025-11-27
>
> ### Concerns and Our Responses
>
> We are grateful for the detailed suggestions. Below, we address common concerns and our new revisions.
>
> ### Scope and framework comprehensiveness – **Clarified**
> - **NFy8** raised concerns about the framework's "comprehensive" claims and connection to real-world use cases.
> - **Clarification**: SteeringSafety is an evaluation framework for representation steering methods, not a normative AI safety framework. Our seven perspectives were selected based on: (1) established benchmarks with validated metrics, (2) prior steering work demonstrating effects on these dimensions, and (3) identification as high-priority risk areas in safety literature like NIST AI RMF and prior work. We clarify that our framework covers perspectives most relevant to current representation steering research where robust evaluation is possible, and will revise language that suggests universal safety coverage.
>
> ### LLM-based evaluation and potential bias – **New Results**
> - **3DPr, cyiT** expressed concerns about LLM-as-a-judge potentially masking biases.
> - **Action**: We selected LLM judges from the original datasets’ papers to ensure consistency. Because all methods are evaluated under identical conditions, our relative comparisons remain robust to systematic biases. We conducted **human annotation studies** to quantify inter-rater reliability between human and LLM judges. Preliminary results validate our LLM-based scoring approach and are included in the revision in Appendix D.2.
>
> ### Additional safety dimensions and low prompt counts – **New Results**
> - **3DPr** requested additional safety dimensions like long-context reasoning; **cyiT, 3DPr** noted low prompt counts for some categories.
> - **New Results**: We have added **evaluation on steered models using LongBench v2** to assess long-context reasoning performance in Appendix Section F.4. We added Table 3 to show the dataset sizes per split and have conducted **statistical significance tests** with appropriate multiple testing corrections for all perspectives to adjust for sample sizes, shown in Figure 6 and onwards.
>
> ### Missing methodological and computational details – **Updated**
> - **3DPr, cyiT** noted missing detail on runtime costs, layer selection, and judge prompting; **T7in** identified confusing mathematical notation.
> - **Action**: We have added comprehensive documentation including: (1) **computational cost analysis** in Appendix Section D.4, (2) **complete methodological appendix** covering layer selection, steering strength, and judge prompting (all replicable in full code release) in Appendix D, (3) **clarified mathematical notation** separating Directional Ablation and Affine Steering into distinct equations in Appendix A.1.3, and (4) **dataset size tables** with split details in Table 3.
>
> ### Dense figures and presentation – **Updated**
> - **3DPr, cyiT** noted that scaled averages and dense figures make results difficult to interpret.
> - **Action**: We have redesigned Section 5, evaluation, with  **more intuitive per-perspective breakdown figures**, **expanded discussion** of effectiveness and entanglement patterns, and **finer-grained analyses** to help interpret safety tradeoffs. Appendix results remain available for granular details.
>
> ### Extension to additional steering perspectives – **New Results**
> - **T7in** noted confusion about restricting to three main perspective axes.
> - **Clarification and New Results**: We originally focused on harmfulness, bias, and hallucination because they align with existing steering literature and have sufficient data for robust evaluation. We have now run additional experiments steering on morality, adding a training and validation set based on additional data from the original source, DecodingTrust. We find morality steering performance is model-dependent and observe counterintuitive results where increasing morality worsens implicit and explicit bias. Full results are depicted in Appendix F.1 and Figures 6, 9, and 12.
>
> ---
>
> ### Summary of New Additions
>
> All new experimental results and rewrites are incorporated into the revised manuscript:
>
> 1. **Human annotation studies** to validate LLM-as-a-judge approach, Appendix D.2
> 2. **Long-context reasoning evaluation** using LongBench v2, Appendix F.4
> 3. **Statistical significance tests** for all perspectives with multiple testing corrections, Figures 6 onwards
> 4. **Extended steering perspectives** beyond original three axes to include morality, Appendix F.1
> 5. **Comprehensive computational and methodological documentation**, Appendix D
> 6. **Redesigned results presentation** with clearer figures and expanded analysis, Section 5
> 7. **Reorganized paper structure** moving Related Work earlier and clarifying terminology, Main Text
>
> We believe these additions substantially strengthen the contribution and address all substantive concerns raised during review. We thank all reviewers again for their valuable feedback that has improved our work.

---

### Meta-Review · Area_Chair_AMzV · 2026-01-10

**Summary:**

There were lost of comments by the reviewers, a few important ones are below:
* Many of the stated contributions in the intro and conclusion are not substantiated by the paper. In addition to the issues of framework comprehensiveness and usefulness mentioned above, there is no effort in the paper to explain why the assessments are "systematic" or "reproducible" or "reliable".
* These are empirical claims about the use of the framework to evaluate real applications; showing that it is possible to pose questions for which evaluations lead to quantitative results only matters if there is meaning to be made from the resulting quantifications.
* The framework identifies safety tradeoffs but does not clearly explain why the studied methods are related to specific harms. The underlying causal mechanisms remain often unclear to the reader.
* causal mechanism is unknown as of now
* The evaluation relies on LLM-based scorers, which can introduce bias and reduce interpretability of the results. Wouldn't evaluation with the help of human annotators for at least a subset of results provide more insights?
* Although the benchmark covers nine safety perspectives, it leaves out some very relevant safety dimensions such as adversarial robustness, long-context reasoning stability, etc.
* For some categories, the number of used prompts seem a bit low.
* There are several misleading/confusing statements in this manuscript that need to be clarified prior to publication. The primary point of confusion is the restriction of measuring steering effectiveness on three main perspective axes (line 67) – it is not evident why only three perspectives are selected and which three perspectives the authors are referring to.

**Reviewer Concerns:**

Though many concerns were addressed by the authors and I appreciate the efforts they put in, the main concerns that I think that still remained open are (also based on my reading of the paper):

* no effort to provide insights into the observations highlighted in the paper
* the paper seems like a concise report of a live code-base that the authors want to put online for the use of the community (which is definitely valuable), but it lacks concrete recipes for practitioners
* several statements are unclear -- a proper revision will be important

**Reviewer Scores:**

It is hard to indicate how the scores would have changed but , after my own reading of the paper, I believe the rebuttal wouldn't have led to a clear acceptance of the paper.

---

### Decision · Program_Chairs · 2026-01-26

Reject